# Assessment of the Genetic Distinctiveness and Uniformity of Pre-Basic Seed Stocks of Italian Ryegrass Varieties

**DOI:** 10.3390/genes13112097

**Published:** 2022-11-11

**Authors:** Elisa Pasquali, Fabio Palumbo, Gianni Barcaccia

**Affiliations:** Laboratory of Genetics and Genomics for Plant Breeding, Department of Agronomy, Food, Natural Resources, Animals and Environment, Campus of Agripolis, University of Padova, 35020 Legnaro, Italy

**Keywords:** SSR markers, Lolium multiflorum, forage crop, DUS test, varietal uniformity

## Abstract

*Lolium multiflorum* Lam., commonly known as Italian ryegrass, is a forage grass mostly valued for its high palatability and digestibility, along with its high productivity. However, Italian ryegrass has an outbreeding nature and therefore has high genetic heterogeneity within each variety. Consequently, the exclusive use of morphological descriptors in the existing varietal identification and registration process based on the Distinctness, Uniformity, and Stability (DUS) test results in an inadequately precise assessment. The primary objective of this work was to effectively test whether the uniformity observed at the phenological level within each population of Italian ryegrass was confirmed at the genetic level through an SSR marker analysis. In this research, using 12 polymorphic SSR loci, we analyzed 672 samples belonging to 14 different Italian ryegrass commercial varieties to determine the pairwise genetic similarity (GS), verified the distribution of genetic diversity within and among varieties, and investigated the population structure. Although the fourteen commercial varieties did not show elevated genetic differentiation, with only 13% of the total variation attributable to among-cultivar genetic variation, when analyzed as a core, each variety constitutes a genetic cluster on its own, resulting in distinct characteristics from the others, except for two varieties. In this way, by combining a genetic tool with the traditional morphological approach, we were able to limit biases linked to the environmental effect of field trials, assessing the real source of diversity among varieties and concretely answering the key requisites of the Plant Variety Protection (PVP) system.

## 1. Introduction

*Lolium multiflorum* Lam., commonly known as Italian ryegrass, is one of the most important groups of grasses largely used as forage crops due to its high productivity and high nutritional value as livestock feed, especially in terms of fiber palatability and digestibility [1,2]. Over the years, advantages have also been demonstrated from an environmental point of view: *L. multiflorum* can be used as a soil stabilizer during the winter to provide ground cover against soil erosion and depletion. Moreover, despite its lower persistence and stress tolerance in comparison with *Lolium perenne*, Italian ryegrass is more productive and can provide faster ground cover due to its timely emergence and seedling vigor.

Both annual and biennial varieties are available on the market, and to prevent any misunderstanding, it is fundamental to clarify that they both belong to this species: the truly annual forms of *L. multiflorum* Lam. var. *westerwoldicum* Wittm. (Westerwolds ryegrass) can be distinguished by their complete flowering in the year of sowing from the biennial forms of *L. multiflorum* Lam. ssp. *italicum* (A. Br.) Volkarts, which generally produce very few seed heads in the sowing year, complete the cycle the following year after cold and short-day conditions, and remain leafy during the entire season.

Although Italian ryegrass is naturally diploid, tetraploid forms have been developed by chromosome doubling to achieve higher biomass production and better nutritional characteristics. The increase in an organism’s cellular ploidy caused by genome replication without mitosis, as occurred in Italian ryegrass using colchicine (i.e., the most efficient anti-mitotic agent and mitosis inhibitor [3]), has been shown to play an important role in physiology and development via cellular, metabolic, and genetic effects [4,5]. Compared to their diploid counterparts, polyploid forage crops display enlarged leaf dimensions and plant height [6,7] but also faster regrowth after grazing and an increasing number of branches and stems [8]. In particular, tetraploid cultivars of Italian ryegrass showed not only faster leaf elongation rates with respect to diploid cultivars, resulting in longer leaves, but also larger shoot dry weights under stress-free conditions [9,10]. Moreover, positive effects have also been reported in enhanced forage quality, since the increased production of secondary metabolites in the induced polyploids together with a higher cell content/cell wall ratio leads to improved succulence and higher forage intake [11,12].

The most common method used to produce Italian ryegrass varieties is based on a recurrent selection strategy, consisting of repeated cycles of intercrossing among selected superior individuals. Almost 700 varieties of Italian ryegrass were reported in the 2022 edition of the OECD (Organization for Economic Co-operation and Development) list of varieties eligible for seed certification [13]. This highlights the economic value of registering a commercial variety and the consequent necessity of protecting breeder rights. The official regulation of Italian ryegrass cultivar identification and registration drafted by the International Union for the Protection of New Varieties of Plants (UPOV) is based on the DUS (Distinctness, Uniformity, and Stability) test, considering all morphological traits, such as length and width of leaves, the intensity of green color, height and width of plants, length of the inflorescence, and the number of spikelets. Ploidy level is the only non-morphological trait considered as a distinction criterion by UPOV. Grow-out tests are performed in both spaced plant and row plot trials, and the visual scoring of traits is frequently performed using numerical scales with different minimum–maximum ranges (e.g., 1–3, 1–5, or 1–9). However, considering the outbreeding nature of Italian ryegrass, and thus the high genetic heterogeneity within each variety, the exclusive use of morphological descriptors to discriminate and compare the increasing number of varieties is becoming difficult and costly. In addition, the morphology and physiology of a plant are the results of strong interactions with environmental factors such as rainfall, temperature, light exposure, and soil composition. Thus, the idea of a unique and fixed guideline based on the evaluation of morpho-physiological traits results in an inaccurate assessment, especially in a worldwide market.

Without a more precise and standardizable characterization, it is impossible to clearly differentiate one variety of Italian ryegrass from another and guarantee real compliance with the DUS requirements in the current Plant Variety Protection (PVP) system.

In this scenario, molecular tools would be useful in implementing the traditional DUS test, as proposed by Gilliland et al. [14], to avoid the plagiarism risk and demonstrate a clear improved value for cultivation and use (VCU) for a new registered variety. Indeed, a DNA marker system could avoid strong environmental dependence, having the potential to investigate many more samples in genotypic assays. In other words, molecular markers such as simple sequence repeats (SSRs) and single-nucleotide polymorphisms (SNPs) represent more robust and objective tools to be integrated into the PVP system for Italian ryegrass. In particular, microsatellite loci consist of short DNA sections of tandemly repeated di-, tri-, and tetranucleotide motifs, and consequent polymorphisms derive from a different number of repetitions. These differences can be detected with conserved PCR primers designed on the nonrepetitive flanking regions. Since they are genetically well-defined and codominant, SSRs have become a powerful genetic tool for marker-assisted breeding (MAS) and the assessment of diversity and uniformity. In addition, SSRs are highly polymorphic markers and are therefore able to discriminate closely related varieties, as often happens for Italian ryegrass. Finally, the SSR assay is easily performed by PCR, a method that does not require specific and expensive laboratory equipment. Despite the recent development of next-generation sequencing (NGS) techniques, particularly restriction site-associated DNA sequencing (RAD-seq) [15] and genotyping-by-sequencing (GBS) technologies, such as genotyping by random amplicon sequencing and direct (GRAS-Di) [16], we want to provide a molecular assay that is less costly and time-consuming and that could be practically applied by small laboratories of seed companies. From the literature, the use of molecular markers in the evaluation of the genetic diversity of *L. multiflorum* accessions has been reported by few studies using sequence-related amplified polymorphisms (SRAPs) [17], randomly amplified polymorphic DNA (RAPD) [18], and SSRs combined with bulk strategies [19].

In this study, we used 12 polymorphic SSR loci to genotype 14 Italian ryegrass commercial varieties, each composed of 48 individuals (672 samples in total). In the downstream analysis, we determined the genetic similarity (GS), verified the distribution of genetic diversity within and among varieties, and investigated the population structure. The main aim was to concretely test whether the uniformity observed at the phenological level within each population of Italian ryegrass was confirmed at the genetic level through an SSR marker analysis. Thus, in this way, we limited biases linked to the environmental effect of field trials. Thus, we were able to evaluate the real source of diversity among varieties and obtained a concrete answer to the fundamental requisites of the PVP system.

## 2. Materials and Methods

### 2.1. Plant Materials

In this study, 14 commercial varieties of Italian ryegrass, differing in ploidy level (diploid or tetraploid) and vegetative habit (annual or biennial) (Table 1), were provided and bred by the same private company. For each variety (here named from A to P), 80 seeds belonging to the latest generation of a purification process were sown in growing boards with a common potting soil and grown in a greenhouse under short-day conditions and a temperature of 20 °C at the “L. Toniolo” experimental farm of the University of Padova in Legnaro (PD). After three weeks, leaf samples were collected from 48 seedlings of each variety, for a total of 672 samples.

### 2.2. DNA Extraction

Genomic DNA (gDNA) was extracted from young freeze-dried leaves finely ground with a TissueLyser II mill (Qiagen, Valencia, CA, USA) using a DNeasy^®^ 96 Plant Kit (Qiagen) following the manufacturer’s protocol. The quantity and quality of the gDNA were checked using a NanoDrop spectrophotometer (Thermo Fisher Scientific, Pittsburgh, PA, USA) in terms of concentration and 260/280 and 260/280 ratios. Extracted gDNA was stored at −20 °C until further PCR amplification.

### 2.3. SSR Marker Genotyping

For genotyping analysis, an initial batch of 28 primer pairs was chosen from Hirata [20] and Guan [21] considering their polymorphism information content (PIC) and their distribution within the seven linkage groups (LG). Preliminary tests on a subset of four randomly chosen samples were performed in singleplex reactions to verify primer efficiency in terms of the presence of polymorphic alleles for each marker. Thus, 12 primer pairs were selected and combined into three different multiplex groups (Table 2) based on similar annealing temperatures, diverse predicted amplicon sizes, and minimum tendencies of dimer formation checked with PerlPrimer v1.1.21. software. Multiplex PCRs were performed according to the three-primer system described by Schuelke [22], with some minor changes. Each locus was amplified by a pair of locus-specific primers, one with an oligonucleotide tail at the 5′ end (M13, PAN-1, PAN-2, or PAN-3) and a third primer complementary to the tail and labeled with a fluorescent dye (6-FAM, VIC, NED, or PET), necessary to subsequently discriminate the different loci during chromatogram screening.

The multiplex PCR mixtures had a total reaction volume of 20 µL resulting from 1 × Platinum^®^ Multiplex PCR Master Mix, 10% GC Enhancer (Applied Biosystems, Carlsbad, CA, USA), 0.25 μM of each tailed primer, 0.75 μM of each tailed reverse primer, 0.5 μM of each fluorescent labeled primer (Applied Biosystems), 20–30 ng of gDNA, and sterile water to volume. Touchdown PCRs were completed for all three multiplexes in 96-well plates under the following conditions: initial denaturation of 5 min at 95 °C, followed by 6 cycles at 95 °C for 30 s, extension at 65 °C for 45 s, which decreased by 1 °C with each cycle, and at 72 °C for 45 s, then 34 cycles at 95 °C for 30 s, at 59 °C for 45 s, and at 72 °C for 45 s. The reaction finished with a final extension step of 30 min at 60 °C. Finally, the quality of the amplicons was verified by electrophoresis run on a 2% agarose/1 TAE gel containing 1× SYBR Safe DNA Gel Stain (Life Technologies).

The PCR products were dried at 65 °C and investigated with capillary electrophoresis using an ABI 3730 DNA analyzer, adopting the GeneScan 500 Liz as the molecular weight internal standard. Thus, chromatograms were screened to define amplicon size at each locus using Peak Scanner software 2.0 (Applied Biosystems).

### 2.4. Genetic Similarity Estimates and Genetic Diversity and Relationship Analyses

Two distinct datasets for marker statistics and analyses were constructed, setting different thresholds of missing data: a dataset admitting samples with missing data of two SSR loci at most, and a second dataset composed of samples with missing data of five SSR loci at most. The strictest dataset (576 samples) was uploaded to the NTSYS software package v.2.21c [23] and used to calculate the genetic similarity (GS) among individuals in all possible pairwise comparisons based on Rohlf’s simple matching (SM) coefficient. This coefficient was used because it also takes into account negative co-occurrences of alleles between different samples and populations. In addition, the average GS within and among each variety was calculated. Using the same software and Rohlf’s GS matrix, a principal coordinates analysis (PCoA) was carried out.

The genetic relationship analysis was performed according to the maximum-likelihood method (ML) implemented in IQ-Tree v1.6.12 software. This approach enabled us to associate phylogenetic and combinatorial optimization techniques into a fast and effective tree search algorithm with the implementation of the ultrafast bootstrap approximation approach [24]. The SSR marker dataset (576 samples) resulting matrix was analyzed as binary data using the GTR2 method (GTR2 + I + G4 + FO), according to the BIC value found with the ModelFinder algorithm available in the IQ-Tree. The GTR model, used to investigate the genetic relationship with SSR data, was also selected according to the scientific research of Huang et al. [25], Vieira et al. [26], and Minin et al. [27]. Statistical support for the ML dendrogram was computed by running 1000 replicates until convergence for ultrafast bootstrap (UFB) (-bb 1000) [28,29] and 1000 rounds of SH-like approximate likelihood ratio tests (SH-aLRT) (-alrt 1000) [30].

On the other hand, the second most comprehensive dataset, including 644 samples, was used both for the SSR statistics and the population structure analysis. GenoDive v3.0 software [31] was used to perform the Analysis of Molecular Variance (AMOVA) through F-statistics to analyze the molecular variance at different levels of the population structure (i.e., individual, subpopulation, and population levels), with the number of permutations = 999. GenoDive allows for a correction of the unknown dosage of alleles, permitting the statistics to estimate without or with a reduced bias. This correction uses a maximum likelihood method based on random mating within populations, as first proposed by De Silva et al. [32] and then modified by Meirmans et al. [33]. Genetic diversity was also investigated through the following statistics: number of observed alleles (N_a_), number of effective alleles (N_e_), and observed (H_o_) and expected (H_e_) heterozygosity within subpopulations assuming Hardy–Weinberg equilibrium according to Nei. These parameters were also calculated for each Italian ryegrass variety.

F-statistics were computed, and the fixation index (F_ST_) and gene flow (N_m_) were calculated for each marker locus. F_ST_ measures the amount of genetic variance that can be explained by population structure based on Wright’s F-statistics [34], while N_m_ = {(1/F_ST_) − 1}/4. An F_ST_ value of 0 indicates no differentiation between the subpopulations, while a value of 1 indicates complete differentiation [35]. Inbreeding coefficients (F_IT_ and F_IS_) were computed to measure the deficiency (positive values) or excess (negative values) of heterozygotes for each assessed microsatellite. Similarly, inbreeding coefficients were calculated at the multilocus level to estimate the genetic effect of total population subdivision as a proportional reduction in overall heterozygosity due to variation in SSR allele frequencies among different subpopulations. Finally, marker allele frequencies for each locus were also determined.

Considering each variety separately, Wright’s inbreeding coefficient G_IS_ was evaluated to investigate the deficiency and excess of heterozygosity within every Italian ryegrass variety. Moreover, a matrix with Nei’s gene diversity (G_st_) values (analogous to F_ST_) was constructed for all possible pairwise comparisons between the fourteen varieties.

### 2.5. Genetic Structure Analyses

In addition, the population structure of the 644 samples was assessed using the clustering algorithm of STRUCTURE v2.2 software. Since no a priori knowledge of the origin of the populations under study was available, the admixture model was chosen. We ran the Markov chain Monte Carlo (MCMC) model with 100,000 iterations and a burn-in of 20,000 samples under the assumption that the allele frequencies in the populations were correlated. Ten iterations were conducted for each value of the number of populations (K), with K ranging from 1 to 20. The obtained results were analyzed using the STRUCTURE HARVESTER web software to calculate the best value of K according to Evanno et al. [36], and then estimates of membership were plotted as a histogram using an Excel spreadsheet.

## 3. Results

### 3.1. Descriptive Statistics of SSR Marker Loci

A total of 239 alleles were detected across the 14 varieties for the 12 microsatellite loci with an average number of observed alleles (N_a_) of 19.9, ranging from 10 (02_01B) to 48 (02_10B). Moreover, the effective number of alleles (N_e_) per locus varied from 1.7 (02_01B) to 5.2 (02_10B), as reported in Table 3. The same parameters calculated for each variety were equal to 7.3 (N_a_) and 3.5 (N_e_), ranging from 4.4 in variety I to 9.3 in variety N and from 2.5 (variety A) to 4.6 (variety D), respectively (Table 4).

Allele frequencies were calculated per locus within each population, permitting the identification of the most common genotype.

Out of 239 alleles, 108 were considered “rare”, since their overall frequency was lower than 1% [37]. Rare alleles were found at each locus, ranging from 4 to 16. Private alleles, those detected only in a specific variety but absent in all the others, were also searched. Fifty-nine private alleles were detected. On average, each variety presented five private alleles (from 1 of A to 20 of O), with the exception of G and I, which did not show any private allele. Most of them, i.e., 52 out of 59, had a frequency lower than 5% but higher than 1% (not considerable as rare alleles).

Descriptive statistics for Nei’s genetic diversity (H-statistics) and Wright’s inbreeding coefficients (F-statistics and G_IS_) for single marker locus and population accessions were computed. The mean observed heterozygosity H_o_ equal to 0.498 ranged from 0.384 to 0.671 among the 12 loci; similarly, H_e_ ranged from 0.42 (02_01B) to 0.827 (02_10B) among loci, with a mean value of 0.649. The inbreeding coefficient (F_IS_) had an average value of 0.302 for microsatellite loci, derived from all positive values, except −0.024 of the 16_03D marker. Finally, F_IT_ and F_ST_ were both positive and, on average, equal to 0.385 and 0.125, respectively, while the gene flow (N_m_) was equal to 1.753. The calculated N_m_ values were >1 in all assayed marker loci (except for 02_02C and 12_05E), ranging from 0.871 to 3.987 over all accessions (on average, N_m_ = 2.183), hence supporting the little genetic differentiation between the fourteen varieties in the analysis.

Considering each variety separately, the mean H_o_ amounted to 0.498, ranging from 0.276 in variety D to 0.686 in variety N, whereas the mean expected heterozygosity H_e_ of 0.652 varied from 0.553 in variety I to 0.727 in variety D. An important finding is that the expected heterozygosity (H_e_) over all the plant accessions scored a mean value significantly higher than the observed heterozygosity. As a consequence, Wright’s inbreeding coefficient G_IS_ scored positive values, revealing a marked deficiency of heterozygotes across more than half of the varieties (especially varieties B, D, E, H, L, and O), although negative values of G_IS_ were estimated for varieties F, I, M, and P, suggesting an excess of heterozygous individuals within these varieties.

Pairwise G*_ST_* values indicate some genetic differentiation between populations (Table 5). Levels of pairwise population differentiation were variable, ranging from 0.019 to 0.236. The lowest G*_ST_* value (0.019) was detected between the M and N varieties, which showed that these varieties had the lowest genetic differentiation and differences. Conversely, the highest G*_ST_* value (0.236) was presented between the A and L varieties, which had the highest genetic differentiation and differences.

Pairwise comparisons among the 12 microsatellites and all 14 varieties revealed a significant deviation from the Hardy–Weinberg equilibrium with a significant multilocus heterozygosity deficiency (Wahlund effect), since at the multilocus level, the inbreeding coefficient G_IS_ showed all positive values, ranging from 0.142 to 0.62 (Table 6).

### 3.2. Genetic Diversity and Clustering Analysis

Genetic variability within and between varieties was investigated primarily by calculating genetic similarity (GS) for all possible pairwise comparisons among the 576 samples (the strictest dataset) using the entire set of marker alleles scored at all genomic loci. In particular, a pairwise genetic similarity matrix was calculated using a simple matching coefficient. Rohlf’s genetic similarity ranged from 78% to 99% among all individuals analyzed. When calculated within each variety, GS varied on average from 88.5% (±2.03) within the N variety to 92.6% (±1.99) within the G variety (Table 7). In the pairwise comparisons between varieties, the N and H populations showed the lowest average value (86.48 ± 1.99%), while the I and A populations exhibited the highest value (90.45 ± 2.07%) (Figure 1).

In addition, an analysis of molecular variance (AMOVA) was conducted to better understand the distribution of genetic differentiation among and within varieties of Italian ryegrass. Findings from AMOVA revealed that 87% (resulting from the sum of 60% found within all the individuals and 27% found among the individuals nested in populations) of the total genetic variation was contributed by differences within varieties, which was notably and significantly higher than that among varieties, since only 13% of the total genetic variation was due to differences among varieties.

Clustering analyses were performed to identify PCoA centroids based on the genetic similarity estimates and to draw the maximum likelihood (ML) dendrogram. Generally, the ML dendrogram revealed a slightly structured distribution of Italian ryegrass accessions (Figure 2). In most cases, individuals from different varieties clustered separately, although two admixed groups were present. Interestingly, the A variety appeared totally distinguishable from all the others, with a statistical UFB support of 79. Also identified were two main monophyletic subclusters, which divided most of the samples of the P-M/N-F-I-L-G varieties from the E-H-D-B-C varieties. Notably, samples from the P and 8 of the D variety formed a distinct cluster that differed from other varieties. Additionally, varieties D, E, and M/N showed slight fragmentation, since they were split into three or two parts. Notably, samples belonging to the O variety did not cluster together, spreading into almost all other clusters, especially into the two admixed groups and with the C and E samples.

Principal coordinates analysis graphically represented the spatial distribution of the samples (Figure 3). To obtain clearer and more readable graphical representations, we plotted the annual varieties separate from the biennial data. In the case of the annual varieties, the first two principal components were able to explain 55.2% of the total genetic variation found within the population as a whole, although most of the accessions were closely plotted in the central area of the four main quadrants. In particular, with the first component, explaining 28.8% of the total diversity, accessions of the I variety could be discriminated from the subgroup samples of O and B and part of the samples belonging to the H variety. The second component, which explained 26.4% of the total diversity, was clearly able to distinguish the F samples from individuals of I and H varieties.

For the biennial varieties, even though most of the accessions were nearly plotted in the central part of the four main quadrants, individuals belonging to the L variety were grouped together and well separated from the other samples. The first two principal components were able to explain 60.3% of the total genetic variation found within the population of the biannual varieties. In particular, the first component, which explained 33.9% of the total diversity, discriminated samples of the L and G varieties from the C and D varieties. On the other hand, with the second component, explaining 26.4% of the total diversity, accessions of the L variety could be distinguished from the G samples and part of the individuals belonging to the M and N varieties. In addition, as shown in the dendrogram, C could not be distinguished from D and partially from E.

### 3.3. Population Structure

Based on the marker alleles at all SSR loci, the genetic structure of the Italian ryegrass core collection (644 samples) was analyzed using STRUCTURE v. 2.2 software [38] and Structure Harvester software [39] to determine the most likely number of ancestral genotypes represented by the core collection. Following the procedure of Evanno et al. [36], a clear maximum for the ΔK value at K = 2 was found (ΔK = 117.5), and a second lowest value was found for K = 13 (ΔK = 9.8). In particular, for K = 2 (Figure 4a), the population was split into two genetically distinguishable subgroups, representing the major marker allele clusters or the ancestral multilocus haplotypes. Similarly, the result observed for K = 13 (Figure 4b) enabled the discriminative clustering of the analyzed varieties, with one exception for varieties M and N.

Starting from the graphical representation of K = 2, each sample is plotted as a vertical histogram divided into K = 2 (and then for K = 13) with colored segments representing the estimated membership in each hypothesized ancestral genotype. The first clustering of the considered genotypes revealed that 518 of the 644 samples showed a strong ancestral association (>90%). Individuals’ membership with the first identified cluster was observed to be higher than 80% in almost all of the analyzed accessions. Particularly, populations A, B, C, D, O, and P had 94%, 91%, 98%, 89%, 92%, and 79% of samples with membership to the first cluster higher than the considered threshold, respectively. On average, all of these varieties presented a membership higher than 90% (except for P with 84%) with the main cluster. On the other hand, populations E, F, G, H, I, and L had 80%, 100%, 100%, 87%, 93%, and 98% of the samples with membership with the second cluster higher than 80%, respectively. Additionally, in this case, the average membership with the second cluster of all these varieties was higher than 90%, with the exception of the E variety with 87%. In contrast, most of the admixed genotypes (<80% membership in a single ancestral genotype) were from the M and N varieties, with 62% and 67% of the samples, respectively. Specifically, these two populations showed complementary average membership to the two hypothesized ancestral genotypes: the M variety had 63% and 37% membership to Cluster 1 and Cluster 2, while the N variety had 40% and 60%, respectively. More generally, other populations had a very low percentage of admixed samples with this clustering, ranging from 24% and 20% for the E and P varieties to the total absence of admixed genotypes in the F and G populations.

The second largest ΔK revealed an additional level of population structure and allowed the clustering of all investigated genotypes into thirteen additional subgroups. Interestingly, the ancestral population size K = 13 corresponded to the number of varieties used in this study, combining the M and N varieties into a unique admixed group. On average, all individuals of the same variety showed a membership higher than 50% to a specific cluster, which was different for each variety (exceptions were observed for varieties M and N). In addition, membership mean values higher than 80% were reached in the B (82.4%-Cluster 12), C (84.6%-Cluster 4), F (84%-Cluster 7), G (83.9%-Cluster 11), I (87.5%-Cluster 1), and L varieties (80.5%-Cluster 13) by the majority of their individuals. The only cluster shared by two different varieties (M and N) was Cluster 3, with an average membership equal to 50.8% and 68.8%, respectively. Moreover, considering those samples with a membership to the major cluster lower than 80% as admixed, the percentage of admixed samples within the same population ranged from 64% (the M variety) to 13% (the I variety). In particular, the highest proportions of admixed samples were shown within the H (58%), D (57%), N (56%), and O varieties (52%). In contrast, the I, C, and F populations had only 13%, 25%, and 25% admixed samples, respectively.

## 4. Discussion

Morphological characteristics have been used for descriptive purposes and are traditionally used to distinguish plant varieties. However, these methods are questionable because of the strong effect of the environment on morphological traits. In addition, this approach is inefficient because of the time and cost involved [40]. Moreover, morphological criteria alone are not sufficient to distinguish some different varieties that are morphologically similar. Therefore, the molecular fingerprinting of a plant variety is extremely important for protecting plant breeders’ rights (PBR) [41,42]. Although some issues were related to the application of molecular tools (e.g., the lack of information related to the allelic dosage, the requirement of an appropriate number of markers, and the need for specific bioinformatic knowledge), SSR markers are considered more reliable than RAPD and AFLP markers because of their ability to produce high-fidelity profiles thanks to their codominant nature and specificity [43]. As a consequence, several researchers have reported the application of SSR markers for genetic diversity assessment and variety identification in crops [44,45,46], horticultural plants [47,48,49], and forages [50,51,52].

In this study, 14 commercial varieties of Italian ryegrass were tested with 12 SSR loci, analyzing a total of 672 samples to assess their intrapopulation and interpopulation genetic variation and differentiation and the structure of the population as a whole, and in turn to evaluate them in the PVP context. Detecting genetic variability by means of marker allele composition, allele proportion, and multilocus genotypes in different accessions of *L. multiflorum* Lam. is a crucial step toward a more trusted variety identification and stronger protection for breeder rights.

As a general rule, the outcrossing breeding system of self-incompatible species, along with the annual life cycle, may have a central role in affecting the conservation of genetic diversity in Italian ryegrass. Indeed, for species with outcrossing mating systems, within-variety variation is usually high because pollen can be widely spread between varieties, resulting in a low degree of differentiation attributable to among-varieties variation [53]. In this study, 87% of the total variation was attributable to within-cultivar genetic variation, while variance among cultivars was only responsible for 13% of the total variation, which is consistent with results from previous studies in cross-pollinated forage plants. For example, Nie et al. [19] reported a within-variety genetic variation value equal to 82% of the total variation, testing six Chinese *L. multiflorum* Lam. varieties. Moreover, in perennial ryegrass cultivars, the total within-cultivar component of genetic variation was 85% [54]. In tetraploid white clover, the within-cultivar variation explained 84% of the total variation [55].

The simultaneous occurrence of a marked inbreeding rate (F_IS_ = 0.302, on average), a reduced fixation index (F_ST_ = 0.125, on average), and a high gene flow (N_m_ = 2.18, on average) between overall accessions confirm that most allele diversity and genotype variation are found within populations and that genetic differentiation among varieties is low. Such a low genetic differentiation can be attributable to a high gene flow among populations because of pollen dispersal among too close multiplication fields of different varieties. In addition, along the variety selection and commercialization process other possible causes could be related to variability of heading time and consequent preferential pairing, climate/location effect during seed batch multiplication, or any possible admixture events during storage and bagging processes. Interestingly, the heterozygosity deficiency seen in our results, with a consequent robust deviation from Hardy–Weinberg equilibrium, is not only a common observation in commercial varieties due to the human-induced bottleneck and artificial selection, but also attests to the low variability in these accessions.

The STRUCTURE analysis suggested moderate structuration of populations and the existence of admixed ancestry based on two or thirteen distinct gene pools. It should be noted that the subdivision in two genetically distinguishable subgroups does not reflect either the two different ploidy levels or the two vegetative habitus. However, the resulting information from the STRUCTURE analysis partially confirmed the outcomes from the maximum likelihood dendrogram. In particular, the I, L, and G varieties could have the same hypothesized ancestral genotype according to STRUCTURE analysis and formed a monophyletic group in the ML cladogram with significant UFB support (91). Moreover, samples belonging to the M and N varieties appear to be closely related to each other. In fact, when K = 13 in the STRUCTURE analysis, M and N samples were the only ones to share the same ancestral gene pool (with the majority of the samples considered admixed, with a membership to Cluster 3 ranging from 79% to 1%), and in the ML dendrogram, they were always mixed and clustered together. Additionally, in the pairwise *F_ST_* comparisons, the M and N varieties showed the lowest value of genetic differentiation (*F_ST_* = 0.019), and again, the intra-variety genetic similarity value of the N population was practically the same as that among the M and N populations (88.54% and 88.59%, respectively). In other words, we can assume that the populations with the highest within variability (M and N), according also to STRUCTURE results, can be considered the most closely related. According to all these results, we can therefore say that the M and N varieties are not distinguishable from each other. In contrast, ML clustering analysis clearly suggested that the A accession has a gene pool clearly distinct from the other accessions, hence clustering apart from the core of varieties. Furthermore, in the pairwise comparisons with other varieties, the genetic similarity values of the L population were the lowest, confirming the resulting values in the pairwise *F_ST_* comparisons.

In conclusion, if it is true that these fourteen commercial varieties of Italian ryegrass did not show high degrees of genetic differentiation, it is also true that, when analyzed as a core, each variety described a genetic cluster on its own, resulting in distinguishability from the others. The only evident exception was represented by the M and N varieties. Given their biennial tetraploid nature, a common recent origin from a shared ancestor could be assumed, as reported also by the population structure analysis. Interestingly, from what was reported in the varietal sheets, these two cultivars are morphologically similar, showing a similar growth habit and comparable heading time and inflorescence length, although the M variety should have longer stems and bigger leaves than the N variety.

The registration of a new variety in the Italian National Register of Varieties still does not take genetic information into account, and the features used to distinguish the candidate variety are genealogical, phenotypic, and other agronomic traits [56]. Molecular characterization by DNA markers, in combination with standard morphological descriptors, can improve variety identification, especially in forage crops. Distinguishing one variety from another is not only the foundation of the seed production market but also the necessary precondition for the successful management of seed companies. In forage crops, especially grasses where morphological differences are not always easily noticeable and objective, the implementation of efficient genetic tools for variety identification is needed. Thus, in the process of potential registration of new varieties, after DUS testing by morphological characters, performing high-throughput marker analysis on the candidate varieties would be highly useful, especially when their phenotypes are not significantly different from existing varieties. This work represents an example of how molecular genetics can be a viable tool to protect both breeders and customers by reducing commercial fraud [57,58].

## Figures and Tables

**Figure 1 genes-13-02097-f001:**
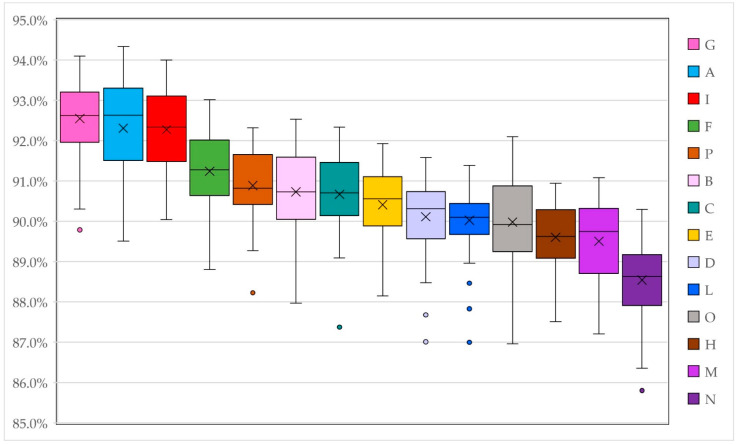
Box plot of mean GS within each variety in descending order. The second and third quartiles are marked inside the square and are divided by a bar (median). The cross (×) within each box represents the mean value. Dots show outlier samples. Different letters and colors refer to the different varieties examined.

**Figure 2 genes-13-02097-f002:**
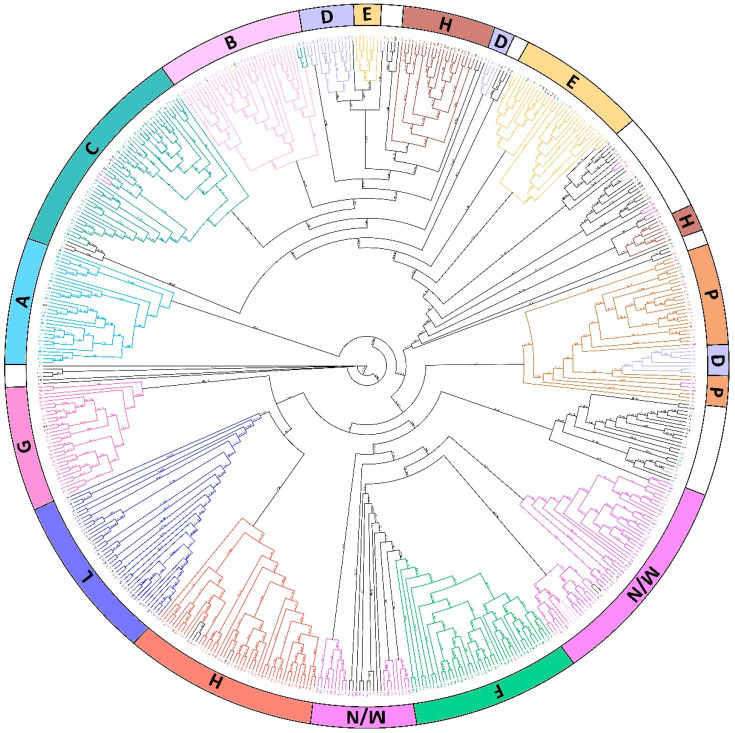
Maximum likelihood dendrogram topology portraying the genetic relationships among the different varieties identified by different colors and letters.

**Figure 3 genes-13-02097-f003:**
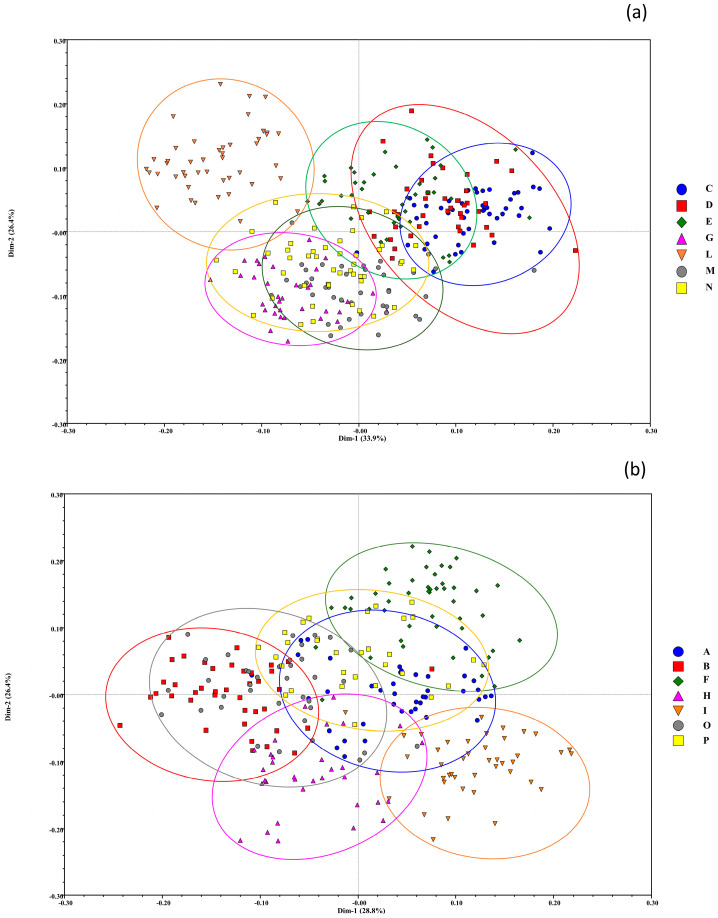
Principal coordinates analysis (PCoA) based on the GS matrix calculated with Rohlf’s coefficient in all possible pairwise comparisons. Samples are labeled following the biotype of belonging (identified by different colors and letters), and their centroids are subgrouped to better represent overlapping areas. (**a**) PCoA of biennial varieties. (**b**) PCoA of annual varieties.

**Figure 4 genes-13-02097-f004:**
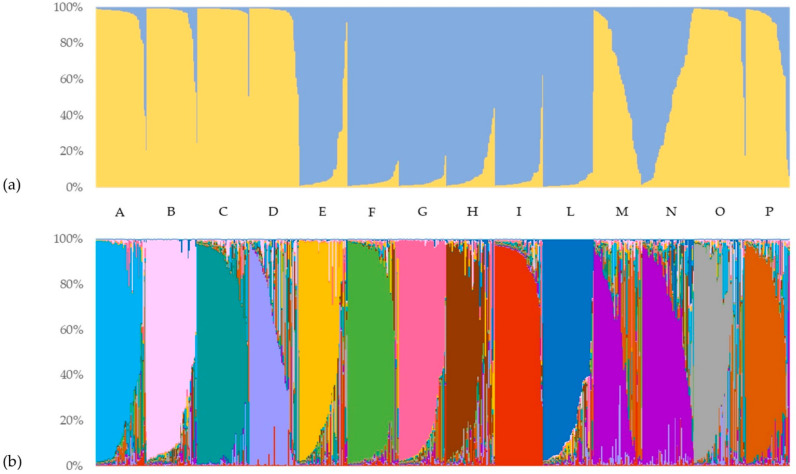
Genetic structure analysis of the Italian ryegrass core collection. Identified most likely value of K = 2 (**a**) and K = 13 (**b**). Clusters and samples’ memberships (indicated by different colors) agree with the variety to which they belong, identified by different letters.

**Table 1 genes-13-02097-t001:** Plant material information, including variety code, relative ploidy level, and vegetative habit.

Variety	Ploidy Level	Vegetative Habit
A	4x	annual
B	2x	annual
C	4x	biennial
D	2x	biennial
E	2x	biennial
F	4x	annual
G	2x	biennial
H	4x	annual
I	4x	annual
L	2x	biennial
M	4x	biennial
N	4x	biennial
O	2x	annual
P	4x	annual

**Table 2 genes-13-02097-t002:** Sequences (5′ to 3′) of the primer pairs used to amplify the SSR markers. For each primer pair, locus name, linkage group membership, maximum and minimum amplicon size (bp), multiplex to which the SSR marker locus belongs, fluorescent dye used, and temperature of melting.

Locus Name	LG	Expected Size	Multiplex	Fluo Dye	Tm (°C)	Forward Primer	Reverse Primer
02-01B	3	309–341	1	6-FAM	60	CTTTATTGCACTTTACTTGCCTTG	CGATGTTCCACGTCAGGTG
13-07A°	2	124–202	1	VIC	62	CACGGAGGCATTTGATTCCC	GCGACCAGTTCCTCGATCT
16-03D	6	153–226	1	NED	63	TTGCTGTTGGGTTTGCTCCC	CTAGAGGTTAGGTCATCTCAAGCG
18-08C	4	317–340	1	NED	62	GCATCAGGGTCGATTCCTC	TTGTCCTATGCTAAAGCTGACATCC
02-10B	6	276–379	1	PET	61	AGTTGGAGTTAACCCCATAGTCAT	CTCATCCATATATAGTCAAGCATAGTG
02-05D	7	243–278	2	VIC	65	ACTGCCTGCACTGGTGCTTG	GCTTACTTCTGCTGACACTGTTTTACA
10-09E	5	204–237	2	NED	62	TCCAAGTGAACGAGTTGCG	TCATCGTCACCACAGTGGC
13-12D	5	285–304	2	PET	63	TTGCTGCTGCACCAATAGCG	GAGCCGATGATGCCACATTC
15-10H	3	200–234	3	6-FAM	62	ACGCTACAAACCGAGGTGG	ATTTGCCTGGAACTGACCCC
12-05E	5	122–160	3	VIC	59	TTCCTGCCGAACGTCTG	CTTGCCAAAAGAGGAGAGGA
14-01A	7	277–318	3	NED	61	CGGCATGCAGTTGAATCTG	ATAATCCACGCCGATCCACG
02-02C	4	180–278	3	PET	61	CCGAATTGTGCCGTATTTGGAT	ACCCACCACATTCGTAAAATG

**Table 3 genes-13-02097-t003:** Descriptive statistics for all SSR loci. The mean number of observed (N_a_) alleles per locus, mean number of effective (N_e_) alleles per locus, observed heterozygosity (H_o_), expected heterozygosity (H_e_), inbreeding coefficients (F_IT_ and F_IS_), fixation index (F_ST_), and gene flow estimates (N_m_) are reported for each SSR locus.

Locus	N_a_	N_e_	H_o_	H_e_	F_IT_	F_IS_	F_ST_	N_m_
02_01B	10	1.7	0.498	0.42	0.074	0.013	0.062	3.917
13_07A	20	2.1	0.384	0.521	0.579	0.517	0.13	1.616
16_03D	14	2.7	0.605	0.636	0.038	−0.024	0.06	3.718
18_08C	15	2.4	0.434	0.596	0.487	0.4	0.146	1.486
02_10B	48	5.2	0.418	0.827	0.605	0.551	0.121	1.644
02_05D	18	4.8	0.671	0.803	0.166	0.114	0.059	3.987
10_09E	18	2.7	0.442	0.639	0.436	0.338	0.148	1.363
13_12D	12	3.6	0.586	0.735	0.251	0.184	0.083	2.691
15_10H	16	3.1	0.462	0.686	0.498	0.42	0.134	1.616
12_05E	21	2.2	0.462	0.545	0.507	0.366	0.223	0.881
14_01A	16	3.1	0.407	0.692	0.641	0.595	0.112	1.962
02_02C	31	3.1	0.602	0.691	0.333	0.144	0.22	0.809
Total	19.9	3.1	0.498	0.649	0.385	0.302	0.125	1.658

**Table 4 genes-13-02097-t004:** Descriptive statistics of genetic diversity calculated for each of the 14 varieties analyzed. The mean number of observed (N_a_) alleles, the mean number of effective (N_e_) alleles, observed heterozygosity (H_o_), expected heterozygosity (H_e_), Wright’s inbreeding coefficient (G_IS_), private alleles (PA) for each variety, and private alleles with a frequency higher than 5% are reported for each variety.

Variety	N_a_	N_e_	H_o_	H_e_	G_IS_	PA	PA > 5%
A	5.8	2.5	0.497	0.571	0.13	1	
B	7.4	3.5	0.3	0.672	0.554	2	2
C	7.3	3.6	0.564	0 664	0.151	5	
D	8.2	4.6	0.276	0.727	0.62	6	1
E	7.3	3.8	0.348	0.682	0.489	6	
F	6.4	2.9	0.648	0 605	−0 070	2	
G	5.6	2.5	0.523	0.562	0.069	-	
H	8.5	4.0	0.424	0.677	0.374	4	1
I	4.4	2.5	0.628	0.553	−0.135	-	
L	8.7	4.6	0.424	0.71	0.402	8	2
M	8.6	4.1	0.68	0.652	−0.043	2	1
N	9.3	4.4	0.686	0.714	0.038	1	
O	7.2	3.9	0.322	0.683	0.528	2	
P	7.9	2.9	0.645	0.616	−0.047	20	
Overall	7.3	3.5	0.498	0.652	0.241	59	7

**Table 5 genes-13-02097-t005:** Matrix with Nei’s gene diversity (G_ST_) values (analogous to F_ST_) for all possible pairwise comparisons between the 14 varieties. Green and red indicate lower and higher G_ST_ values, respectively.

**A**	--													
**B**	0.135	--												
**C**	0.136	0.079	--											
**D**	0.129	0.077	0.055	--										
**E**	0.172	0.121	0.12	0.09	--									
**F**	0.152	0.139	0.135	0.13	0.093	--								
**G**	0.17	0.202	0.184	0.185	0.19	0.174	--							
**H**	0.186	0.145	0.115	0.082	0.11	0.159	0.162	--						
**I**	0.106	0.167	0.16	0.158	0.143	0.15	0.15	0.126	--					
**L**	0.236	0.205	0.208	0.179	0.169	0.202	0.175	0.12	0.167	--				
**M**	0.102	0.097	0.099	0.066	0.105	0.095	0.128	0.134	0.138	0.174	--			
**N**	0.11	0.098	0.102	0.073	0.107	0.096	0.139	0.121	0.133	0.13	0.019	--		
**O**	0.093	0.099	0.076	0.067	0.108	0.133	0.159	0.124	0.126	0.184	0.069	0.081	--	
**P**	0.116	0.125	0.097	0.094	0.145	0.117	0.13	0.141	0.158	0.195	0.054	0.069	0.086	--
	**A**	**B**	**C**	**D**	**E**	**F**	**G**	**H**	**I**	**L**	**M**	**N**	**O**	**P**

**Table 6 genes-13-02097-t006:** Inbreeding coefficient G_IS_ in pairwise comparisons among the 12 SSR loci and the 14 varieties under study. G_IS_ values at the multilocus level are also reported.

Population	02_01B	13_07A	16_03D	18_08C	02_10B	02_05D	10_09E	13_12D	15_10H	12_05E	14_01A	02_02C	Multilocus
A	−0.042	0.165	−0.293	0.606	0.539	0.081	0.527	0.038	0.501	0.616	0.537	0.561	0.317
B	0.301	0.801	0.555	0.386	0.730	0.492	0.603	0.134	0.810	0.455	0.883	0.507	0.554
C	−0.177	0.693	−0.243	0.394	0.396	0.268	0.555	1.000	0.460	0.835	0.209	0.431	0.405
D	0.416	0.701	0.400	0.827	0.531	0.512	0.762	0.858	0.470	0.643	0.831	0.438	0.620
E	0.067	0.714	0.323	0.939	0.773	0.189	0.465	0.322	0.649	0.299	0.636	0.336	0.489
F	−0.062	0.392	−0.427	0.423	0.446	0.070	0.063	0.035	0.443	0.023	0.697	−0.126	0.144
G	−0.102	0.607	−0.404	−0.125	0.560	0.083	0.612	−0.115	0.168	0.181	0.357	−0.241	0.161
H	0.140	0.901	0.286	0.554	0.584	0.485	0.560	0.152	0.799	0.516	0.452	−0.138	0.450
I	0.153	0.112	−0.298	0.624	0.747	−0.087	−0.180	−0.210	0.794	0.370	0.266	−0.288	0.142
L	0.287	0.808	0.474	0.275	0.547	0.319	0.135	0.507	0.423	−0.008	0.752	−0.191	0.402
M	−0.071	0.360	−0.201	−0.032	0.683	−0.041	0.236	−0.019	0.180	0.410	0.555	−0.082	0.163
N	−0.169	0.666	−0.128	0.131	0.420	−0.108	0.345	0.095	0.337	0.368	0.663	−0.015	0.216
O	0.103	0.387	0.533	0.901	0.899	0.054	0.689	0.563	0.525	0.445	0.816	0.427	0.538
P	−0.154	0.198	−0.328	−0.157	0.561	−0.042	0.131	−0.016	0.035	0.264	0.918	0.463	0.172
Overall	0.067	0.542	0.033	0.417	0.596	0.160	0.388	0.243	0.464	0.392	0.612	0.193	0.348

**Table 7 genes-13-02097-t007:** Average genetic similarity percentages (GS%) calculated within and among each variety using Rohlf’s simple matching coefficient. Red and green indicate lower and higher GS%, respectively.

**A**	92.31%													
**B**	89.42%	90.73%												
**C**	89.28%	89.28%	90.67%											
**D**	89.12%	89.21%	89.28%	90.11%										
**E**	88.83%	88.70%	88.66%	88.85%	90.41%									
**F**	89.31%	88.16%	88.02%	88.01%	89.30%	91.24%								
**G**	90.29%	88.19%	88.50%	88.17%	88.58%	89.35%	92.55%							
**H**	88.08%	87.87%	88.03%	88.42%	88.44%	87.52%	88.72%	89.60%						
**I**	90.45%	88.38%	88.26%	88.07%	88.83%	89.16%	90.35%	88.99%	92.28%					
**L**	87.25%	87.09%	86.61%	87.02%	87.56%	86.86%	88.22%	87.96%	88.34%	90.02%				
**M**	89.31%	87.89%	87.99%	88.08%	88.15%	88.69%	88.97%	86.86%	88.56%	86.70%	89.50%			
**N**	88.60%	87.53%	87.24%	87.50%	87.63%	88.10%	87.99%	86.48%	88.04%	86.82%	88.59%	88.54%		
**O**	89.64%	88.99%	88.97%	88.95%	88.64%	88.27%	88.69%	87.87%	88.46%	87.15%	88.33%	87.66%	89.98%	
**P**	89.92%	88.27%	88.76%	88.43%	87.98%	88.71%	89.92%	87.29%	88.96%	86.71%	89.24%	88.37%	88.87%	90.88%
	**A**	**B**	**C**	**D**	**E**	**F**	**G**	**H**	**I**	**L**	**M**	**N**	**O**	**P**

## Data Availability

The data are contained within the article and supplementary materials are available on request from the corresponding author. The raw data are not publicly available due to privacy reasons.

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
