# Peer review of "Assessment of the Genetic Distinctiveness and Uniformity of Pre-Basic Seed Stocks of Italian Ryegrass Varieties"

_genes, 2022, doi:10.3390/genes13112097_

Round 1

Reviewer 1 Report

Interesting study showing some potental of using molecular markers in DUS test.

Some remarks:

Line 10-14: correct:  However, Italian ryegrass has an outbreeding nature and therefore has high genetic heterogeneity within each variety. Thus Consequently, the exclusive use of morphological descriptors in the existing varietal identification and registration process based on the Distinctness, Uniformity, and Stability (DUS) test results in is an inadequately precise assessment.

Line 16: correct: In this research, by Using 12 polymorphic SSR loci, we analyzed 672 samples belonging to….

Line 22: add: except for 2 varieties

Line 31: Do you have a reference for fiber palatability of Italian ryegrass? Because of stem formation digestibility of Italian ryegrass is often lower than other grass species.

Line 53: Some of the mentioned advantages of tetraploids are not general and not applicable to Italian ryegrass (e.g. number of stems). I suggest to focus on differences between diploid and tetraploid Italian ryegrass.

Line 55: succulence: Tetraploid Italian ryegrass has a lower dry matter content than diploid Italian ryegrass because of the bigger cells and as a consequence a higher cell content/cell wall ratio.

Line 65: What do you mean by “except for ploidy level”? Ploidy level is a distinction criterion for UPOV

Line 98: Correct: time- and money-consuming that and could be practically applied by small laboratories

Line 103: Correct: In this study, by using  we used 12 polymorphic SSR loci, we  to genotyped 14 Italian ryegrass..

Line 104: why 48 individuals? In DUS testing 60 individuals are required.

Line 404: Would you recommend to use also agronomic behaviors to distinguish varieties. Please comment.

Line 462-467: If you would apply only molecular markers for DUS testing, than all 14 varieties would be distinct except M and N. Were there any morphological (or agronomic) characteristics  to distinguish M and N?

Author Response

Line 10-14: correct:  However, Italian ryegrass has an outbreeding nature and therefore has high genetic heterogeneity within each variety. Thus Consequently, the exclusive use of morphological descriptors in the existing varietal identification and registration process based on the Distinctness, Uniformity, and Stability (DUS) test results in is an inadequately precise assessment.

Done, thank you

Line 16: correct: In this research, by Using 12 polymorphic SSR loci, we analyzed 672 samples belonging to….

Changed

Line 22: add: except for 2 varieties

Added

Line 31: Do you have a reference for fiber palatability of Italian ryegrass? Because of stem formation digestibility of Italian ryegrass is often lower than other grass species.

References added

Line 53: Some of the mentioned advantages of tetraploids are not general and not applicable to Italian ryegrass (e.g. number of stems). I suggest to focus on differences between diploid and tetraploid Italian ryegrass.

Thank you for your remark, a more focused sentence was added

Line 55: succulence: Tetraploid Italian ryegrass has a lower dry matter content than diploid Italian ryegrass because of the bigger cells and as a consequence a higher cell content/cell wall ratio.

We included this information, thank you

Line 65: What do you mean by “except for ploidy level”? Ploidy level is a distinction criterion for UPOV

You are right, but we meant that all morphological traits and just one non-morphological trait (namely ploidy level) are considered in UPOV evaluation. Anyway, we rephrase this paragraph for a better clarity.   

Line 98: Correct: time- and money-consuming that and could be practically applied by small laboratories

Done

Line 103: Correct: In this study, by using  we used 12 polymorphic SSR loci, we  to genotyped 14 Italian ryegrass..

Corrected

Line 104: why 48 individuals? In DUS testing 60 individuals are required.

You are right. While these cultivars have been already tested for DUS criteria by the competent body using the number of samples you were referring to (60), the aim of this study was to further investigate a slightly smaller set of samples (48) through molecular analysis. This choice (48) was made basically for technical reasons, namely to fit two lines (48 samples x 2) in each 96well-plate.

Line 404: Would you recommend to use also agronomic behaviors to distinguish varieties. Please comment.

With that sentence we meant to highlight that morphological criteria alone are not sufficient to distinguish different varieties that are morphologically very similar. To avoid any misunderstanding, we modified this part.

Line 462-467: If you would apply only molecular markers for DUS testing, than all 14 varieties would be distinct except M and N. Were there any morphological (or agronomic) characteristics  to distinguish M and N?

Thank you for this interesting remark, we integrated this part with some information reported in the official descriptive sheets of these two varieties to underline also some morphological similarities and differences. 

Reviewer 2 Report

Dear Authors

I found the manuscript well done and extensively deepened in the frame of a PhD work; the following remarks and/or questions are only in the aim of improving your thesis defence as well as for the interest of the readers of the journal:

Background of DUS:

-        I guess that all your 14 varieties met standard DUS requirements but could you tell more about them? Do explicit their commercial names and year of registration into the EU list; were all varieties bred by the same breeding company or independently, you just mentioned “a private company”? Are there pairs of varieties more or less close in respect with DUS test?

-        What do you mean in the title by “pre-basic seed stocks”, first generation after polycrossing? just before commercial release? As DUS control in the grasses is performed at commercial generation not before; it may make the issue of your work questionable it comparison of marker distinctness is not performed at the same generation as DUS.

-        Possibly, your manuscript would have been even more convincing by using true candidates cultivars having NOT met DUS requirements but YES in respect with VCU ones.

-        Consider more generally the paper referring to the potential use of a molecular DUS by T. Gilliland et al.: a proposal for enhanced EU herbage VCU and DUS testing procedures, DOI: 10.1111/gfs.12492, in Grassland and Forage Science.

SSR Markers

-        The total number of alleles is extremely high for SSRs, 239, of which, almost half is under frequency p < 0.01, of which half again appear to be variety-specific (rather than “private”) alleles; do you have explanation for this? May capillary electrophoresis introduce artefacts vs gel electrophoresis? Does multiplexing PCR amplification add just insignificant noise? Is the distribution of allele frequency consistent with number of SSR repeats? In any case, you could have confounded those variety-specific alleles with the closest ones in term of number of repeats to get more robust mean allele frequencies; anyway, did-you find, at least, some trigenic or tetragenic genotypes involving rare alleles within the tetraploid varieties?

-        The fact that allelic dosage was not possible is a strong limitation in your work, in particular in respect with the use of F-statistics and Nei’s indicators which require fully informative genotyping;

Even if you tell that diploid and tetraploid varieties were not distinguishable by SSR, it may be quite interesting that you showed in a table with all mean indicators you computed between diploids and tetraploids. It’s a little strange by some way that molecular marker are found to be unable to distinguish between diploids and tetraploid while morphological traits of DUS did it.

In the discussion, you could add some words on alternative present methods of genotyping by sequencing (SNP) on various issues, costs, allelic dosage, number of markers, needs in bioinformatics, ...

-        give, if known, the linkage groups on which the 12 SSRs are located

Methods

-        I can understand that you overview many methodologies of assessing genetic differentiation for publication sake but many are redundant, inadequate or poorly explained by some instances: what is the Rohlf’s simple matching (SM) coefficient? The IQ-Tree v1.6.12 software procedure (lines 173-182) remains quite obscure as written.

-        My opinion is that F-statistics (and Nei’s one) are not sensitive enough for molecular discrimination between varieties and are only interesting for quoting past studies in bred varieties of allogamous species or wild population of the same species (ie grasses). I can remember that Fst in natural populations of L. perenne may range from 0.05 to 0.10 so that mean Fst of 0.125 between varieties here is not so much more.

-        The GS without any statistics (ie Student test of mean GS within variety) seems also even less sensitive than Fst; are table 7 and fig 1 actually useful? If you keep them anyway, use the same colour code for table 5 and table 7 to appreciate, at least visually, how Gst and GS correlate; you could use a Mantel test (see in the bibliography) of correlation between matrix.

-        I found quite effective that Structure software detected possible admixture within the M and N varieties although I did not understand how M variety has cluster 3 membership of 50.8 % while % of admixed samples is 64 %, same with N variety with 68.8 % and 56 % respectively. % of admixed samples remains unclear to me.

-        Comparison between Structure results and dendrogram in fig2 is difficult to understand if not actually explained; in addition to variety O, the variety I is not displayed in fig 2. Also, why the M and N varieties are confounded into a unique M/N class in the fig? Alternatively, you could draw a dendrogram by using individual cluster membership of each variety after Structure, K=13, this would help any comparison with fig 2.

-        You used PCoA but why not Factorial Discriminant Analysis and testing how many individuals are well classified into each variety. You could also withdraw from linear combination of allele frequencies and correlations with the factors, which SSR are the most effective for robust a posteriori classification. Robust dendrograms could be also drawn simply from Euclidian distance using the coordinates of individuals on the first factors.

-        Generally speaking, a logical presentation of results would be to deal first with within variability, and then to deal with variability between varieties of a similar within variability. The case of your varieties M and N illustrates that populations having the highest within variability (Structure results) may come to the conclusion that they are the most closely related (Fst of 0,09).

Discussion

-        Carefully interpret Fis as an inbreeding rate since it considerably overestimates F calculated from genealogy. The number of entries of each variety, if known, gives far better estimates.

-        You restrict low differentiation to contamination among seed multiplication fields while there are many other effects within and between varieties from the initial polycross up to the commercial seeds 3 or 4 generations later: preferential pairing due to variability of heading date among entries, unbalanced seed yield; climate/location effect across seed batch production, … as well as all possible admixture events in the course of bagging process.

-        I suggest that you conclude your manuscript by referring to a simple and unique dendrogram, at variety level, which would give in a glance how the 14 varieties are close as a whole and which molecular indicator appears relevant enough, according to you, to be used as a threshold value for distinctiveness. This could be of great value for any official testing in the EU and to document the introduction of a molecular DUS at UPOV.

Author Response

Background of DUS:

Thank you for all your precious observations. For privacy reasons, many details about the varieties examined have been omitted, such as commercial names and year of registration in EU list, but we have nevertheless tried, where possible, to integrate some of the information requested. In this case, we specified the common breeding origin and the fact that are commercial variety which have undergone a varietal purification process to acquire greater morphological uniformity and stability over generations.

In the title we used “pre-basic seed stocks” referring to the last generation seed of the previous mentioned purification plan of already commercial varieties. For a better clarity and to prevent any misunderstanding, we added this information in the 2.1 section. Moreover, we found very interesting the paper of Gilliland et al. you suggested us, since it could make our proposal more actionable by placing it in a broader context in which the use of molecular markers is also associated with the assessment of VCU. Therefore, we cited this interesting work in the introduction.

SSR Markers

First of all, we want to thank you for your accurate remarks. As regards the considerable amount of alleles found and the fact that quite half of them showed a low frequency, we can affirm that capillary electrophoresis is very sensible and able to distinguish alleles that differ by 1 bp, differently from a gel electrophoresis. As specified in the 2.3 section, some random samples were tested in Singleplex PCR and showed the same pattern later found in Multiplex PCR. Those aspects led us to exclude the possibility that artefacts could be produced in Multiplex PCR or that some noises were generated due to the use of capillary electrophoresis.

As concern rare and private alleles, it is not always true that all private alleles resulted rare too, since the majority have a frequency lower than 5% but higher than 1%, within the specific variety. We added some details within the text for a better clarity. Anyway, if we understand correctly your comment, we verified the presence of rare alleles in triallelic genotypes and we found for example two rare alleles of two different markers that are involved in triallelic genotypes in two different tetraploid varieties together with more frequent alleles. This could represent a confirmation of the effective existence of the detected rare alleles in our populations.

As regards the difficulties encountered in distinguishing diploid and tetraploid accessions based on SSR markers, we do agree that it may sound little strange. However, our experience along with the scientific literature (e.g. “Genetic Diversity in Switchgrass Collections Assessed by EST-SSR Markers” by Narasimhamoorthy et al. 2008 and “Genetic Diversity of Populations of Saccharum spontaneum with Different Ploidy Levels Using SSR Molecular Markers” by Liu et al. 2015) would confirm the hypothesis that SSR are not useful to discriminate different levels of ploidy. In addition, the inability to assess allelic dosage could be considered an additional obstacle in distinguishing diploids from tetraploids with SSR markers. The most adequate method to distinguish different ploidy levels still remain flow cytometry, together with the traditional chromosome counting.  

Thank you for these last two remarks, we integrated the discussion pointing out some of the possible problems related to the use of molecular tools in DUS test that have to be considered from the perspective of practical application.

Since linkage group membership was one of the factors considered in the initial choice of marker loci from the previous studies, we added this important information in Table 2.

Methods

Our intention was to try to evaluate genetic differentiation with more approaches, since we expected high rates given the allogamous nature and total self-incompatibility of ryegrass. As requested, we implemented the materials and methods, specifying which parameters are considered in calculating the the Rohlf’s simple matching (SM) coefficient and adding some additional information about IQ-Tree software.

As suggested, we unified colour code among GS (Table 7) and Gst matrices (Table 5) to make their correlation clearer and more obvious. Now for instance, in both matrices, L variety line is characterized by red colour corresponding to low genetic similarity (GS) rates with other varieties and, on the other hand, elevate Nei’s gene diversity (Gst) values in the pairwise comparisons with other varieties.

Moreover, for Structure results, to increase the clarity, we integrated both the results and the discussion sections specifying what we meant for “admixed samples”, namely those samples with a membership to the major cluster lower than 80%. In this way, for example, the percentage of membership to cluster 3 of M variety (50.8%) was not in contrast with the rate of admixed samples (64%), since in the first case we refer to the mean value of the membership to the main identified cluster, whereas with the second we are just giving the percentage of samples showing a not so high (lower than 80%) membership.

In figure 2, varieties O and I are not displayed because their samples are scattered throughout the ML cladogram, especially in those sections in black that don't have any labels, so we preferred to not draw them since they did not form defined clusters. On the other hand, M and N samples are displayed as a unique class for greater graphic clarity since their samples resulted always clustered together. In addition, for each variety we used the same colour in both ML cladogram and Structure histograms, to highlight the correspondences between the results of the two different analyses.

Thank you for the suggestion about M and N varieties, we integrated the discussion following your last remark, trying focusing readers’ attention to the possible correlation between different analyses results even in this case.

Discussion

Thank you again for the suggestion about Fis, we considered it as an inbreeding coefficient in accordance with various works from the literature, such as “SSR-Based Analysis of Genetic Diversity and Structure of Sweet Cherry (Prunus avium L.) from 19 Countries in Europe” by Barreneche et al. (2021), “Selection signatures across seven decades of hard winter wheat breeding in the Great Plains of the United States” by Ayalew et al. (2020) and “Genetic diversity in old populations of sessile oak from Calabria assessed by nuclear and chloroplast SSR” by Lupini et al. (2019).

We added some other factors that could lead to such a low genetic differentiation during all the entire selective and productive process. As you suggested, indeed, we pointed out the possibility of admixture events related to a not so careful management of storage and bagging processes, which could be quite dangerous especially in a very large company such as the one that supplied the varieties under consideration.

I want to thank you very much for your time, thoroughness, interest, and for all the food for thoughts you gave me. These not only helped me improve (as much as possible) the manuscript, but also gave me valuable insights for my PhD discussion. Moreover, I make myself available to further clarify certain aspects should you find it appropriate.